# Interventions for Maintenance of Essential Health Service Delivery during the COVID-19 Response in Uganda, between March 2020 and April 2021

**DOI:** 10.3390/ijerph191912522

**Published:** 2022-09-30

**Authors:** Steven Ndugwa Kabwama, Rhoda K. Wanyenze, Suzanne N. Kiwanuka, Alice Namale, Rawlance Ndejjo, Fred Monje, William Wang, Siobhan Lazenby, Susan Kizito, Christopher Troeger, Anne Liu, Helena Lindgren, Neda Razaz, John Ssenkusu, William Sambisa, Rebecca Bartlein, Tobias Alfvén

**Affiliations:** 1Department of Community Health and Behavioral Sciences, School of Public Health, Makerere University, Kampala P.O. Box 7062, Uganda; 2Department of Global Public Health, Karolinska Institutet, 17177 Stockholm, Sweden; 3Department of Disease Control and Environmental Health, School of Public Health, Makerere University, Kampala P.O. Box 7072, Uganda; 4Department of Health Policy Planning and Management, School of Public Health, Makerere University, Kampala P.O. Box 7072, Uganda; 5School of Public Health, Makerere University, Kampala P.O. Box 7072, Uganda; 6Gates Ventures, Kirkland, WA 98033, USA; 7Department of Women’s and Children’s Health, Karolinska Institutet, 17177 Stockholm, Sweden; 8Department of Medicine, Karolinska Institutet, 17177 Stockholm, Sweden; 9Department of Epidemiology and Biostatistics, School of Public Health, Makerere University, Kampala P.O. Box 7072, Uganda; 10Bill and Melinda Gates Foundation, Seattle, WA 98109, USA

**Keywords:** COVID-19, health care, learning health systems, health services, public health

## Abstract

Introduction: The COVID-19 pandemic overwhelmed health systems globally and affected the delivery of health services. We conducted a study in Uganda to describe the interventions adopted to maintain the delivery of other health services. Methods: We reviewed documents and interviewed 21 key informants. Thematic analysis was conducted to identify themes using the World Health Organization health system building blocks as a guiding framework. Results: Governance strategies included the establishment of coordination committees and the development and dissemination of guidelines. Infrastructure and commodity strategies included the review of drug supply plans and allowing emergency orders. Workforce strategies included the provision of infection prevention and control equipment, recruitment and provision of incentives. Service delivery modifications included the designation of facilities for COVID-19 management, patient self-management, dispensing drugs for longer periods and the leveraging community patient networks to distribute medicines. However, multi-month drug dispensing led to drug stock-outs while community drug distribution was associated with stigma. Conclusions: Health service maintenance during emergencies requires coordination to harness existing health system investments. The essential services continuity committee coordinated efforts to maintain services and should remain a critical element of emergency response. Self-management and leveraging patient networks should address stigma to support service continuity in similar settings and strengthen service delivery beyond the pandemic.

## 1. Introduction

The COVID-19 pandemic has overwhelmed both public and private sector health systems all over the world, impacting the delivery of essential health services [1,2,3]. Health system challenges during the COVID-19 pandemic have been attributed to the failure of countries to comprehensively implement global health policy approaches, specifically the Global Health Security (GHS) and Universal Health Coverage (UHC) [4]. GHS includes the proactive and reactive investments to minimize the risk and impact of acute public health events that endanger the health of people within and across geographical regions and international boundaries [5]. On the other hand, UHC focuses on access to health care with minimal financial burden on patients. Low- and middle-income countries (LMIC) have disproportionately and differentially adopted and implemented the two approaches [4]. In addition, the public health and social measures that LMICs have adopted have impacted on health care access and delivery [1,3]. For example, COVID-19-related movement restrictions affected access to immunization services in Pakistan [6], mental health and gender-based violence services in Bangladesh, Kenya, Nigeria and Pakistan [7]. In Uganda, the 2019/2020 health sector performance report noted a 30% drop in ART refills for people living with HIV in the period April–June 2020 compared with January–March 2020 [8]. Thus, health care systems were not resilient and failed to control the epidemic while rendering essential health services simultaneously. Resilient health systems involve a complex combination of interdependent interactions of health actors, institutions and populations, with the wherewithal to resist, prepare for and effectively control public health emergencies, maintain core functions during the emergencies and learn from them to transform and improve the system where necessary [9]. A robust response would, therefore, require a policy mix that allows for a balance between the containment of the spread of disease and the maintenance of essential service delivery. This has called for the development of innovations across the building blocks of a health system [10] to promote the continuity of access to care.

At the onset of the COVID-19, the World Health Organization (WHO) provided member countries with general operational guidance on the maintenance of service delivery including modifications to the prevention, diagnosis, management and treatment of diseases [11]. The guidance proposed indicators for monitoring health services continuity as well as practical recommendations of strategies to be implemented at various levels of the health system to maintain access to safe and quality health services. However, the adaptation and implementation fidelity of the WHO guidance was heterogenous and varied across countries depending on health system capacity and structure as well as local contexts [12]. Responses to previous public health emergencies have been characterized by significant disruption of other essential health services. We conducted this study to describe the interventions instituted to minimize the impact of the pandemic on the health system and maintain the delivery of essential health services in order to inform the current and future responses to public health emergencies as well as health system strengthening.

## 2. Methods

### 2.1. Study Design

We conducted a qualitative study that involved a review of documents on interventions implemented for health services maintenance and interviews with key informants at various levels of the health system, including from national to community level.

### 2.2. Description of the Health System in Uganda

Uganda’s health system is composed of the Ministry of Health (MoH) at the national level as well as the district health services. The MoH oversees health service delivery at the 5 national referral hospitals and the 17 regional referral hospitals and is responsible for policy formulation and analysis, strategic planning, setting standards and quality assurance, resource mobilization, capacity development, technical support supervision, coordination of health services and research, and monitoring and evaluation. At district level, there are 136 districts. The district health structures oversee the health service delivery within their jurisdiction including the planning and implementation of human resources for health policies, recruitment and management of human resources. Health services at district level are provided by both the public and private sub-sectors. The public sector includes the District General Hospitals, Health Centers with increasing levels of service specialty from IIs, IIIs and IVs, and community health workers/village health teams.

### 2.3. Maintenance of Essential Health Service Delivery

In March 2020, the WHO provided operational guidance for the maintenance of essential services during the COVID-19 pandemic [13]. The guidance described maintenance of essential services as one of the strategic shifts to ensure that populations obtain maximum benefits from health reasources that had become increasingly limited due to the COVID-19 pandemic. The guidance included the strategic planning and coordinated action to mitigate the risk of system collapse and ensure continued access to services. The interventions described in this paper include strategies that were implemented to ensure continued and uniterrupted delivery of services.

## 3. Data Collection

### 3.1. Document Review

Several documents and secondary data sources were reviewed to obtain information on the interventions and types of modifications adopted within the health system for health services maintenance. The documents reviewed included the Uganda COVID-19 Interventions Report 2019/2020 [14], the Uganda Annual Health Sector Performance Report 2019/2020 [8], the Uganda Annual Health Sector Performance Report 2020/2021 [15], the Uganda Guidelines for Continuity of Essential Health Services [16], the Uganda COVID-19 Preparedness and Response Plan 2020/2021 [17], the Uganda COVID-19 Resurgence Plan 2021/2022 [18] and other grey literature published about the impact of COVID-19 during the peak period between March 2020 and April 2021. Findings about the type of interventions and modifications adopted within the health system to ensure continued access to essential health services were summarized into an online template and categorized according to the WHO building blocks of a health system [10]. The WHO building blocks of a health system include leadership and governance, service delivery or service provision, financial resources, health workforce, health information systems and access to essential medicines. The study modifed the WHO “access to essential medicines” classification with the category “health infrastructure and supplies” as described by Van Olmen et al. [19] to encompass investments in infrastructure, equipment and any other commodities for the maintenance of essential health service delivery.

### 3.2. Qualitative Data Collection

Twenty-one (21) key informant interviews (KIIs) were conducted to document innovations and interventions adopted for the maintenance of essential health services delivery. A key informant interview (KII) guide was used to elicit information on interventions related to apriori themes based on the WHO building blocks of a health system [10]. Key informants included national level policy makers such as members of the national committee on continuity of health services, directors/commissioners in charge of health and clinical services at regional level, district health officers, health facility staff such as nurses, midwives and community health workers. Key informants at national level provided perspectives relating to the strategic, administrative and policy interventions to maintain service delivery. Key informants at district and health facility level provided perspectives related to the the operational adjustments to maintain service delivery. The data collected through KIIs supplemented findings from the document review.

### 3.3. Qualitative Data Analysis

All interviews were audio recorded and then transcribed verbatim. We conducted thematic analysis [20] by carefully reading the transcripts to analyse, interprete and identify codes which were then grouped into themes according to the building blocks of a health system [10]. Key informant quotes from the transcripts are presented to support findings for each of the identified themes.

### 3.4. Ethical Considerations

This study was part of a multi-country project that assessed the response to the COVID-19 pandemic in sub-Saharan Africa [21]. We obtained ethical approval from the Makerere University School of Public Health Higher Degrees Research and Ethics Committee (Protocol #903) and registered the study with the Uganda National Council for Science and Technology (Approval #HS 1121ES). All key informants provided informed consent before participating in the study.

## 4. Results

### 4.1. Essential Health Services during the COVID-19 Pandemic

The Uganda Ministry of Health prioritised specific health services based on the health sector development plan and the extent to which the disruption of the health services would negatively impact health outcomes generally and affect the response to the COVID-19 pandemic [16]. The prioritised essential health services included those related to health promotion and disease prevention (outreaches, referrals, diagnostic services, etc.), prevention and management of communicable (malaria, HIV/AIDS, TB, vaccine preventable diseases, etc.) and non-communicable diseases (hypertension, diabetes, cardiovascular diseases, etc.) as well as maternal, child and adolescent health services (antenatal care, newborne care, postnatal care, immunization services, etc.).

### 4.2. Interventions to Maintain Delivery of Essential Health Services

The Uganda Ministry of Health (MoH) in partnership with local and international actors implemented several interventions at national and subnational level to ensure maintenance of access to essential health services in the country. Figure 1 summarizes the types of interventions implemented at various levels of the health system in Uganda for maintaining access to health services.

### 4.3. Leadership, Governance and Coordination

The interventions for health service maintenance related to leadership and governance included the establishment of coordination structures at national and subnational levels, development and dissemination of guidelines for service continuity, engaging of private sector (both health and non-health), to support efforts to minimize service disruption and communicating to the public about the continued availability of health services in facilities.

(a)Establishment of national committee on continuity of essential health services. In April 2020, the Uganda MoH constituted a committee at national level to ensure the continuity of provision of essential health services [22]. Members of the committee included MoH officials, representatives from District Local Governments, national public health authorities, Health Development Partner Organizations such as UNICEF and WHO, and other Civil Society Organizations. The committee was chaired by the Director of Clinical Services who was the National Focal Point for essential health services continuity. The committee coordinated all efforts to promote maintenance of access to health services. District Health Officers were invited to present to the committee about the trends in access to care within their jurisdictions. Where disruptions were noted, interventions were proposed and implemented.(b)Provision of guidance on health services continuity. In April 2020, the MoH published guidelines for the continuity of essential health services [16]. The guidelines defined the priority essential health services and provided guidance on immediate actions for the continuity of health access and monitoring service delivery. In June 2021, the MoH published a COVID-19 resurgence plan that was informed by lessons learnt in the first year of the response to the pandemic [18]. The plan strengthened the coordination of the COVID-19 response through the establishment of support teams for essential health services maintenance at regional level, occupational health and safety in health facilities and maintaining access to essential medicines and commodities.(c)Coordination for health service continuity at subnational and facility levels: At subnational level-regional, district and facility levels, the government established sub-committees to provide oversight and coordinate the maintenance of essential health services in the context of COVID-19 [16,22]. With respect to the functionality of these sub-committees a key informant noted.


*“We had meetings and decided to divide ourselves in such a way that there are those on the frontline in running COVID-19 activities in the hospital and other colleagues would continue running routine medical services in the hospital and the region at large. These meetings and coordination were not just done at the hospital level, but we also had the district on board, the District Health Officer (DHO) together with his team, and the Resident District Commissioner (RDC)*
*”*
KII Two Regional Referral Hospital

(d)Risk communication and health promotion. The MoH also developed information, education and communication (IEC) materials and conducted media campaigns encouraging people to continue to access health services while preventing COVID-19 [15,22] and combating stigmatization of persons recovering from COVID-19. Health facilities used multi-media such as television and radio to communicate to the public about procedures for accessing emergency and other health care as noted by the key informant below:


*“… (the hospital) started its first use of call and dispatch center for the ambulance during the COVID-19 time…. We disseminated these numbers on radio and … were receiving calls from the communities on those numbers to go and pick stranded patients using our ambulances….”*
KII Eight, Regional Referral Hospital

There were also efforts by specific disease programs to promote the maintenance of essential health services. For example, at the National Tuberculosis (TB) and Leprosy Program, a TB Implementing Partners’ coordination mechanism supported community involvement and awareness about COVID-19 to address the reduction in general outpatient department attendance of HIV/TB patients at health facilities following the COVID-19 outbreak. The mechanism also addressed out of stock GeneXpert cartridges and non-testing of TB samples by laboratory personnel due to lack of personal protective equipment [8].

(e)Private sector engagement. By including the private sector in the COVID-19 response effort at both national and district level, this category of stakeholders was able to contribute to the response and support continuity of services. They provided financial support, personal protective equipment and transport for both patients and health workers. For example, at the national level, at least 65 motor vehicles and 19 motorcycles were received from the private sector as in-kind donations to support both the COVID-19 response [15] and the maintenance of access to health services. Key informants acknowledged receipt of such support from the private sector.


*“We received a few private donors to give us items that were used in the management of COVID-19 at that time including things like masks, face shields, we received things like money, there is an organization… which gave us seventeen million shillings (5000 USD). We were able to buy a washing machine using that money, we were able to make a few repairs before the ministry money came in…a few repairs of the place… mattresses, blankets… basins... soap.”*
KII 9 Regional Referral Hospital


*“(private organizations) gave us means of transport. Like the vehicles I was speaking about. We had several organisations. Apart from the district vehicle, at least we had a vehicle from World Vision, we had a vehicle from Save the Children, we had a vehicle from World Harvest Mission helping in ferrying our staff and Clients to and from hospital. I’m talking about organizations bringing in physical materials to do with infection prevention. (they also) gave us gloves, they gave us chlorine powder, they gave us items really to use during the outbreak.”*
KII 6 District Hospital

### 4.4. Health Workforce

The health workforce interventions to promote health services maintenance involved the promotion of health and safety of health workers, maximizing the use of available staff through re-deployment, provision of incentives and engaging community health workers in the provision of specific services.

(a)Development of guidelines and protocols for health worker safety. The MOH developed guidelines for managing health care workers who contracted COVID-19 while on duty. The guidance covered issues related to routine screening, limiting the entry of COVID-19 exposed caregivers at health facilities, use of personal protective equipment and promoting hand hygiene [23]. Furthermore, the MOH published a health facility screening algorithm to aid regular COVID-19 screening among health workers to avoid infection [16], promote their safety and facilitate maintenance of service delivery.(b)Utilization of e-Platforms for Support Supervision, Capacity Building and Telemedicine: Support supervision and mentorship to improve health service delivery was provided via e-platforms or using telephone during the period under study [16,22]. For example, implementing partners transitioned from face-to-face to online training to build the capacity of health workers to provide family planning services during the COVID-19 pandemic. During financial year 2020/2021, health workers from 135 districts were trained in surveillance, contact tracing and provision of home-based care among others [15] via e-platforms in order to decongest health facilities. Furthermore, health workers leveraged e-platforms, tele-pharmacies and tele-laboratories to provide services such as triage, referrals and mobile medical services [24]. The challenge with the use of technology during the COVID-19 response was that the cost of internet remained prohibitively high and internet services coverage were mostly available to the urban population around the country’s capital Kampala city [24].(c)Re-deployment and recruitment. Several districts in the country recruited and/or re-deployed staff to other facilities to maintain services while a few were assigned to provide care at COVID-19 facilities. In addition, more than 500 health workers were recruited at various levels of the health system to support the COVID-19 response [15]. This contributed to the slight improvement in staffing capacity in health facilities from 73% in 2019/2020 [8] to 74% in 2020/2021 [15].(d)Provision of financial and non-financial incentives to health workers. Furthermore, the MoH adopted strategies that supported health workers to continue providing health services. These included the provision of financial [14,22] and non-financial incentives such as accommodation and transport. At the peak of movement restrictions, health workers were transported to health facilities that were understaffed to fill human resource gaps [22]. This was noted by key informants as follows:


*“…we would move the doctor to do caesarean section especially if the patient was in a facility where the theatre was operational, rather than bring them to the regional referral hospital…. so we take the doctor there and bring them back when they are done.*
*”*
KII One, District Health Officer

Additionally, psychosocial support was provided to infected and affected staff as well as the MOH instituted risk communication targeting health care workers.


*“I can also say there were a lot of efforts on psychosocial support, we had a lot of discussions talking to them, encouraging them, counselling staff and motivating them to work by staying at duty. And at some point, we informed people for instance people who worked so much had to take off time to rest*
*”*
KII Ten, Senior Medical Officer

However, some key informants noted that the allowances were either provided late or were inequitably distributed among the health workers.

(e)Engaging community health workers: Community health workers continued to support the provision of health services at community level during the COVID-19 period under review [16]. The National Malaria Control Program provided community health workers with infection prevention and control commodities and was able to ensure maintenance of community-based services such as indoor residual spraying for mosquito control and integrated community case management of childhood illnesses during the COVID-19 pandemic [8]. Furthermore, the role of community health workers was emphasized in the key informant interviews:


*“We asked (the community health workers) to move within the communities where they are to help and mobilize people to come and access care in the hospital. …we would work with them to mobilize HIV+ clients in their communities so that they would meet at a central place in a particular school or church, then they would go there with their pills and then the pills would be given to the patients that they have mobilized in their region.”*
KII Two, Regional Referral Hospital

For example, Living Goods, a non-governmental organization that supports community health workers by leveraging technology provided 4300 community health workers with personal protective equipment [25]. The organization developed a mobile phone application that community health workers uploaded to their smartphones. This application facilitated adherence to the MOH guidance on preventing COVID-19 as they undertook their responsibility for providing care and treatment for diarrhea and malaria for children under 5 years and supporting mothers with antenatal and postnatal care needs.

### 4.5. Provision of Financial Resources

The interventions related to financial resources included the mobilization of funds from government and international agencies to support service maintenance as well as the continued provision of government financial resources to health facilities to aid the provision of services.

(a)Supplementary funding for COVID-19 response from government and international agencies. The parliament of Uganda approved a supplementary budget amounting to 30.7 million USD (114 billion UGX) towards the COVID-19 pandemic response [8]. This translated to about 4.1% of Uganda’s health sector budget in the financial year 2019/2020 [15]. The resources supported all aspects of the response including the payment of contact tracers, the procurement of diagnostic test kits and the strengthening the capacity of intensive care unit (ICU) in the country. In addition, international development organizations such as UNICEF, Global Fund and USAID provided resources to support health services maintenance [8,15]. According to the MoH COVID-19 resurgence plan (June 2021–July 2022) [18], 31 million USD was budgeted to finance the activities related to the maintenance of essential services during the period. The activities included supporting the national medical stores to maintain access to essential commodities, strengthen reporting and monitoring of service delivery through tracking performance using standardized indicators.(b)Continuation of recurrent funding by government. Health facilities continued to receive the quarterly disbursements of funds for the implementation of all essential health services which minimized health services disruption as noted by a key informant below:


*“We normally get funds for primary health care and we continued to get it, there was no shortage of funds, the funding was as it used to be.”*
KII Nine, General Hospital

### 4.6. Infrastructure and Commodities

Interventions related to infrastructure and commodities included the promotion of safety in health facilities through provision of personal protective equipment. The uninterrupted supply of commodities was achieved through reviewing supply plans, making provision for emergency orders and allowing inter-warehouse tranfers of health commodities.

(a)Provision of infection prevention and control commodities. The Environmental Health Department of the MoH procured and distributed commodities that promoted infection prevention and control in health care settings including hand washing facilities, hand sanitizers and handheld sprayers to 941 health care facilities in 44 districts [8,14].(b)Review of Supply plans. The pharmacy division of the MoH reviewed the supply plans for antiretroviral drugs (ARVs), commodities for voluntary medical male circumcision, drugs for treating opportunistic infections, reproductive health and laboratory commodities to avoid stockouts [8]. This enabled the maintenance of delivery of drugs and medicines for essential services as noted by a key informant:


**
*“*
**
*As for the supply of medicine and other supplies, we maintained coordination with National and Joint Medical Stores (drug distributors), we would make our order and they would deliver in time.*
*”*
KII Seven, General Hospital

(c)Emergency orders and inter-warehouse transfers of essential commodities. Where commodities and supplies ran out of stock, the national distribution mechanism allowed for emergency orders [8,22] as noted by a key informant below:


*“…I remember an emergency order was made to purchase some personal protective equipment (PPEs) because in our hospital we never had masks… aprons were not enough and other PPEs so emergency order was done to purchase that equipment…”*
KII Three, General Hospital

In addition, there was inter-warehouse transfers of medicines such as ARVs and reproductive health commodities across the different warehouses located across the health system tiers to mitigate shortages.

### 4.7. Health Service Delivery

The adaptations to service delivery to promote service maintenance involved the management of COVID-19 cases at higher levels of the health system which have intensive care capacities. This permitted the maintenance of essential services at other levels, promoted self-management and minimized the need for health facility visits. Services in community settings were bridged through multi-month drug dispensing and leveraging existing patient networks to provide medicines.

(a)Designation of facilities for COVID-19 treatment. Uganda’s health system is composed of the national referral hospitals, regional referral hospitals, District Hospitals, Health Center IVs, IIIs, IIs and community health workers. The system is referral based with more complex and specialized services offered at higher levels of the health system. The regional referral hospitals were designated as COVID-19 treatment units so that health service delivery at other levels of the health system could continue [14,16]. However, the designation of the regional referral hospitals for managing COVID-19 patients also affected access to other non-COVID-19 services as noted by an informant below:


*…the (patients) had fear of being in hospital environments. Some of them had the fear that they could contract COVID-19 from hospitals especially they started learning that we had admitted patients. Some of them feared to come to the hospital for that reason...”*
KII Eight, Regional Referral Hospital

To address the challenges of fear, the MoH developed health messages informing the public about the continued availability of other services in lower-level health facilities and regional referral hospitals, in addition to the COVID-19 treatment and management [15,22].

(b)Promotion of self-management. The AIDS Control Program promoted HIV self-testing through development and dissemination of HIV self-testing videos and brochures in multiple local languages [8].(c)Provision of services in community settings: Health workers conducted targeted and integrated antenatal care and immunization outreaches within communities to extend services especially in hard-to-reach communities with many pregnant women [16,22]. As noted by several key informants, health facilities engaged in various activities to take services to communities to ensure maintenance of service provision:


*“We continued to offer immunization outreaches, we continued to offer drug distribution especially for HIV and those ones who were having hypertension, diabetes. We would move out after announcing then we find these people in the communities and deliver the medicine to them”*
KII Ten, General Hospital

(d)Dispensing medicines for multiple months. In addition, patients with chronic disease conditions, especially those with HIV, were given medicines for 3 or more months to decongest health facilities, minimize transmission of COVID-19 and to protect people with underlying conditions [8,15,22].


*“We have also learnt that for stable patients you do not need to see them weekly, monthly, we have been able to maintain the mode of giving them drugs for 3 to 6 months and that way we have been able to decongest even the HIV clinics and then we have been able to focus on those that are failing treatment and those who have challenges so that we offer them quality care*
*”*
KII Eight, Regional Referral Hospital

However, dispensing drugs for multiple months was associated with some challenges. For example, there were medicines out of stock because patients did not obtain the drugs from the facilities they normally received them as a result of movement restrictions imposed during the period under review [26].

(e)Leveraging patient networks to deliver medicines. For disease conditions such as HIV where there was an existing network where patients within specific communities knew each other, service providers gave medicines to one patient who then distributed them to patients within their community network [8,15]. This was also noted by key informants:


*“I think the other innovation is that of grouping the HIV clients and asking them to pick their medicines and one person comes and picks and takes it to them at given point and distributes.*
*”*
District Health Officer

However, some patients were uncomfortable with community distribution of medicines because they did not want other community members knowing their HIV status [26].

### 4.8. Health Information Systems

(a)Provision of reporting tools. To address the challenge of timely reporting into the national surveillance system, the Division of Health Information engaged the District Health Officers and district biostatisticians to step in for the complete and timely submission of service statistics and reports to the relevant platforms. In addition, the MoH provided information and communication hardware including phones, tablets and computers to 135 districts to improve timely reporting [15]. Other interventions to address reporting gaps were issuance of circulars and provision of transport to reporting personnel as noted by two key informants:


*“We sent circulars to the members to make sure that we continued with our reporting and surveillance, we encouraged a lot of reports, that’s at a health facility ward level for example every morning the in-charges they had to tell us.”*
KII Three, General Hospital

## 5. Discussion

The COVID-19 pandemic demonstrated how public health systems with already limited capacity became overwhelmed. The Government of Uganda demonstrated that several innovations, adjustments and adaptations within the health system were implemented to maintain health service delivery across the spectrum of the building blocks of a health system. Given the complex nature of the health system and the interdepencies across the building blocks, there was simultaneous and parallel adoption of innovations across the health system to sustain performance and ensure that the system absorbed the shocks generated by the pandemic.

First, committees were established at multiple levels to coordinate mechanisms for all efforts to maintain service delivery. In February 2020, the WHO published a strategic preparedness and response plan [27] that was equivocal about which strategies could be implemented to ensure essential services maintenance. By March 2020, the first operational guidance on the continuity of service delivery was published [13]. This recommended the establishment of governance and coordination mechanisms for service delivery maintenance in addition to other response protocols. Although the interruption of service delivery during responses to public health emergencies has been previously reported [28,29,30], the need to take deliberate strategies, such as establishing coordination structures for service maintenance, has come to the fore during the response to the COVID-19 pandemic. In fact, monitoring the continuity of service delivery was not included in the standard operating procedures for the coordination of public health emergency preparedness and response [31]. The establishment of the committee underlined the importance of coordinating efforts to ensure that services are maintained as response operations are conducted. The committee identified and prioritized essential health services in the context of the disease burden in Uganda. It provided a platform for multiple stakeholder engagement to develop interventions that minimized the disruption of access to services. In sum, the coordination of health service maintenance at the various levels of the health system should remain a critical element of any public health emergency response.

Across the building blocks of the health system, alterations to standard practice were mainly related to the health workforce and health service delivery. Health workforce modifications were mainly related to increasing numbers through recruitment, redeployment, reassignment and use of other health facility personnel to support the response. These human resource for health modifications and alterations exposed the prevailing shortage of health workforce for mounting a robust response to public health emergencies in Uganda. The significance of sufficient human resources for public health emergency preparedness and response cannot be overemphasised and has been highlighted by existing Global Health Security capacity definitions [32,33] and assessments [34,35]. However, there is a variation in the scope and definition of the human resources required for public health emergency response whereby the WHO Joint External Evaluation [34] and the Global Health Security Index [35] focus on the size of the public health workforce particularly epidemiologists, while the Health System Resilience Index [33] and the Epidemic Preparedness Index [32] have a broader definition that includes district, health facility and community staff. There is a need to broaden the scope of the human resource for emergency preparedness to include other medical workforces such as physicians and nurses, and public health workforce including epidemiologists and community health workers. These issues should be considered in the development of comprehensive national strategic plans for the improvement of human resources for response and control of public health emergencies. In addition to increasing the number of health workers, the provision of financial and non-financial incentives for health workers during epidemics is beneficial in not only motivating them to continue working but in preserving their lives during response to public health emergencies. This has also been noted by researchers elsewhere who noted that community health workers who were paid and consistently supervised were effective in the maintenance of community based materanal and child health services [36]. During Uganda’s response, the provision of transport enabled health workers to overcome barriers of transportation imposed by the lockdown. It also reduced their transportation costs during the pandemic. Financial incentives boosted their morale for work, although these were not evenly spread across the workforce. This study highlights the gap and critical value of prioritizing the health and safety of health workers during response to public health emergencies.

The other set of interventions was related to alterations and modifications of health service delivery some of which might have utility for improving patient care and management during and beyond the COVID-19 pandemic. These modifications included the initiation of self-management, expansion of supply chains and delivery systems for TB and HIV drugs and dispensing drugs for longer periods, the leveraging technology to conduct capacity building and leveraging patient networks for drug distribution. All these interventions intended to minimize the risk of COVID-19 transmission in health facilities by reducing the number of hospital visits without negatively impacting the quality of care. Empowering patients to engage in self-management practices such as HIV self-testing has proven to be cost-effective when implemented in a setting where the prevalence of the undiagnosed population is over 3% [37]. Dispensing medicines for longer periods also alleviates disruptions in access to care, has the potential to include previously excluded populations [38] and is not inferior to standard care in terms of retaining patients in care [39]. Although patient networks were used to deliver medicines for patients with chronic conditions, there were challenges related to stigma which should be addressed before scaling up beyond the COVID-19 pandemic. Other considerations for the success of community models of care include the importance of sufficient stocks of drugs, access to quality clinical care and a reliable network of community health workers [26,40]. However, future research should establish whether the strategy can be implemented without compromising the quality of care for the patient.

Furthermore, another important action taken by the Uganda government and which could have a positive impact on access and continuance of provision of health servives was the leveraging technology for training of service providers, provide supportive supervision and maintain access to laboratory and pharmacy services during the COVID-19 response. All these could strengthen the delivery of primary health care and reduce disparities in health access in the long term [41]. The unintended impacts of service delivery re-adjustements such as re-designation of facilities and rescheduling of service delivery times on access to other services, such as maternal health and non-communicable diseases including mental health, are also worth monitoring in future epidemics.

## 6. Limitations

This research describes the interventions, adaptations and innovations for the maintenance of essential health services but does not provide information on the effectiveness of these changes, which was beyond the scope of this work. However, the merits, limitations and considerations of scaling up the interventions have been highlighted, citing other work performed in similar settings. Secondly, a lot of information is provided from the health provider and district leadership perspective because we did not obtain information from the demand side or health user perspective which would have enriched this research. Future research would benefit from investigating these perspectives to generate a holistic view about strategies for health service maintenance during emergencies.

## 7. Conclusions

Investment in strong health systems before crises occur can mitigate the disruption to service delivery when the crises do occur. Health systems need to adopt innovative strategies to ensure uninterrupted access to health services. The interventions to ensure maintenance of health service delivery in this study include innovations that have utility for future public health emergencies. The establishment of coordination structures for health service maintenance at various levels of the health system should be a critical element of public health emergency response, and the development of health service maintenance plans should be integrated into preparedness efforts. Modifications of standard practice such as self-management and dispensing drugs for longer periods, leveraging technology for training and service delivery as well as leveraging patient networks can also support service maintenance in similar settings and strengthen service delivery beyond the COVID-19 pandemic.

## Figures and Tables

**Figure 1 ijerph-19-12522-f001:**
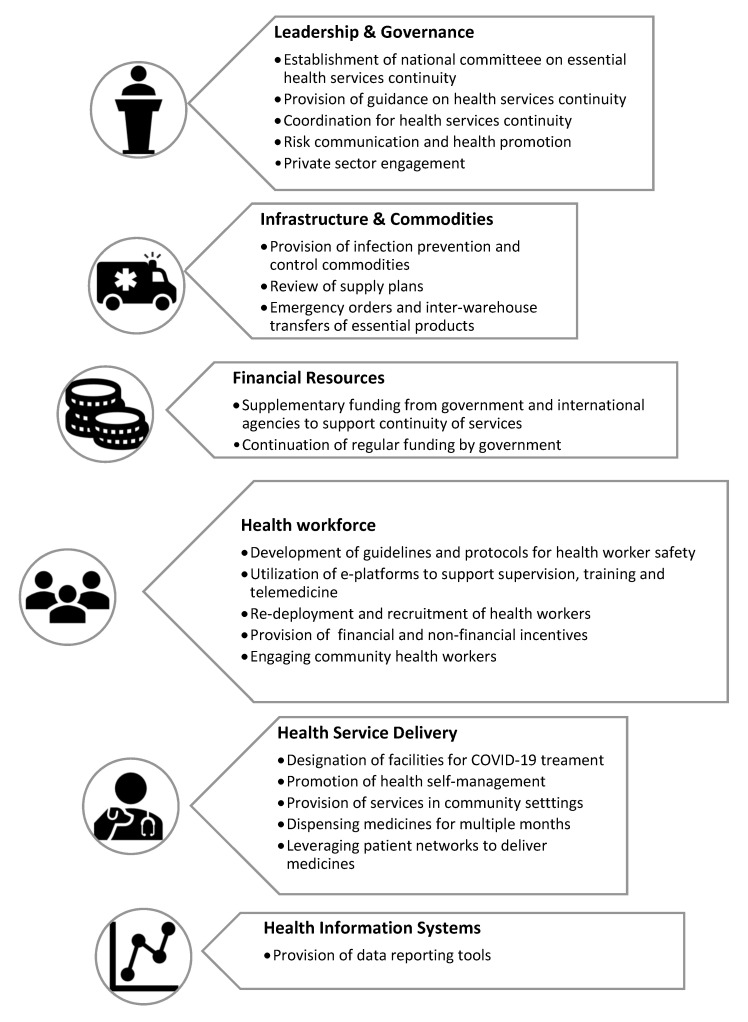
Interventions Implemented to Promote Continuity of Essential Health Services categorized by the WHO Building Blocks of a Health System between March 2020 and April 2021, Uganda.

## Data Availability

The dataset used for analysis can be availed upon reasonable request by writing an email to the corresponding author.

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
