# Peer review of "Interventions for Maintenance of Essential Health Service Delivery during the COVID-19 Response in Uganda, between March 2020 and April 2021"

_ijerph, 2022, doi:10.3390/ijerph191912522_

Round 1
Author Response
Dear Dr. Cecilia Xing
We thank the reviewers for expeditiously reviewing our paper and giving us valuable comments to help us improve its quality. Below this message are point by point responses to all the comments that were raised by each of the reviewers. We thank you for considering our paper in your esteemed journal. I have submitted the revised version of the paper on behalf of my co-authors. All changes to the original version of the paper have been left in track-changes so that the reviewers can easily see them.
Regards,
Steven Ndugwa Kabwama MSc Public Health
Reviewer #1
For authors.
Comment: Introductory remarks.
The notion "continuity of care" appeared in the theory and practice of health care in the 1950s in the patient-provider context. In the 1970s, it became more complicated because continuity meant time relations and coordinated uninterrupted care. Since the 1990s, continuity has been increasingly seen as a multidimensional concept. Today still is a problem with the definition. These two words have different meanings. We have a couple of different "continuities'. Moreover, they often have shown significant overlap with related concepts. Therefore it creates difficulties in measurement. On the other hand, the common opinion is that continuity of care has been identified as an essential element of adequately organised and functioning health care systems.
Continuity of care is the factor, among others, of a better quality of care, better patient satisfaction, reduced hospital use and even mortality, adherence to medications, and more significant equity and satisfaction of patients. Defining what we will do is worthwhile when dealing with the continuity of care. What kind of continuity we are talking about?
Response: We agree with the reviewer about the various ways in which continuity of care can be defined and understood. From the comment of the reviewer, they understood continuity of health services (and rightly so) as the quality of care experienced by the patient and delivered by the provider over time as described by Gulliford et al. However, the overarching message of the paper was intended to be the interventions for the uninterrupted delivery of services when health systems are faced with shocks like the COVID-19 pandemic. While continuity of care entails for example the availabilility of HIV testing services, followed by counselling for those found HIV positive and then initiation of treatment and availability of ARVs as noted by Gulliford et al., this was not the focus of the paper. We aimed to describe interventions that ensured that testing services, counselling services, drugs remained available in circumstances where the health system was faced with a shock. Therefore, we have reworded the title and content of the paper from “continuity” to “maintenance” of service delivery (Lines 1-3). In addition, we have added a paragraph to describe maintenance of services as it was defined by WHO in the operational guidance to promote uninterrupted delivery of services during the COVID-19 pandemic (Lines 158-166).
Comment: There is also no single definition of essential health services. Each country has its list that differs based on a couple of factors changing over time because of socio-economic and political situations and sometimes social values. Universal health coverage is a concept and movement/ being also a dream /to reach a situation where all individuals and communities receive the health services they need without suffering financial hardship.
Response: We agree with the reviewer. The definition of which services are essential and need to be prioritized is context dependent. We have added a definition of essential services and the rationale that was used to prioritize the services in Uganda (Lines 211-220).
Comment: The Global Health Security Index assesses and benchmarks health security and related capabilities that comprise the State's Parties to the International Health Regulations. But despite the general recognition of the social importance of health security, there is no single standard definition of this concept. The same is as regards the meaning of resilience in health systems / or resilient health care system- Int.J.Environ.Res.Pub.Health,2022,19/. Infectious disease outbreaks periodically threaten the human beings population. As well as other disasters. I can't help saying that it would be much easier to control when a critical notion is agreed upon. Hence, we must often clear the decks to avoid misunderstandings and thoroughly study public health issues. It should be done, especially when international journals are concerned.
Response: We agree with the reviewer about including a definition of health security as it applies to the paper. Global Health Security encompasses the proactive and reactive investments to minimize the risk and impact of acute public health events that endanger the health of people within and across geographical regions and international boundaries. This definition has been included in the paper (Lines 106-109). The definition of resilience as it applies to a health system had already been included (Lines 119-122).
The paper.
Comment: Steven N.Kabwama et his team prepared the paper titled " Interventions for continuity of essential health service delivery during COVID-19 response in Uganda, between March 2020 and April 2021". It looks exciting at first sight because of the well-known global situation of curtailment the essential health services during the COVID-19 pandemic. How to fight against coronavirus having the health care system work without additional drawbacks to people's health could be a fascinating story. I want to say a few words about the structure of the paper with some comments on the content. I was referring to the most critical issues, which made me doubtful. And at the end, I will offer more general propositions.
Response: We thank the reviewer for this positive response and the suggestions that have been made to improve the quality of the paper.
Comment: In part of the paper called " introduction", there is some general information on the COVID-19 pandemic's impact on delivering essential health services; some examples and explanations. I am not much of an expert on the English language, but I would like to ask you to give the text to the native speaker. Considering the problem under discussion, the issue of polish and easy-to-read language may not be decisive. But the text is difficult to read. Much more difficult because of the manner of writing and complicated wording than because of the complex essence of the issue. Example?. Voila! "....The health system response to the pandemic was characterised by the a lack of resilience of the system across its various tires exacerbated by the lack of preparedness and failures to simultaneously control the spread of COVID-19 while ensuring the continuity of essential routine service delivery." Uff! What about like this, straightforward:
" The health care systems were not resilient against the COVID-19 pandemic. One of its features was its failure to control the epidemic and to render the essential health services simultaneously". Or something like that.
Or more "seriously": "The health system response to the pandemic was characterised by lack of resilience across its various failures to simultaneously control the spread of COVID-19 while ensuring the continuity of essential routine service delivery".
Response: We thank the reviewer for raising these comments – we appreciate the broad readership of the journal and the importance of communicating in a clear and succinct manner. We have reworded this particular sentence (lines 117-119) as suggested by the reviewer and had the paper re-read and reviewed to improve the sentence construction, grammer and flow of the entire manuscript.
This part of the paper also describes the aim/s/ of the study. I have understood that there are three main aims. Namely:
- description of the interventions instituted to minimise the impact of the pandemic on the health system,
- promotion of the continuity of access to non-COVID -19 health services to inform the current and future response to public health emergencies. This sentence does not sound very clear, especially as regards the relations between continuity and information. And also between "non-COVID 19 healh services" and "COVID -19 services/?/.
Response: We appreciate the comment. We have clarified that the study aimed to describe the interventions instituted to minimize the impact of the pandemic on the health system and maintain the delivery of essential health services.
- provision of intelligence towards health system strengthening and health service continuity efforts to buttress the shock of future public health emergencies. I am pretty sure the authors were not thinking about "buttress the shock" but about "butress of pandemic control"; increasing the strength of the answer to the shock, not the shock itself. Anyway, it is the next example of style problems.
Response: We thank the reviewer for catching this. We have removed the confusing language and clarified that the study aimed to describe the interventions instituted to minimize the impact of the pandemic on the health system and maintain the delivery of essential health services and inform the current and future response to public health emergencies as well as health system strengthening (lines 136-139).
Comment: Part two of the paper is devoted to the description of the methods. The first sentence of the part named"Study design" informs readers that "it is a qualitative study with a review of documents on interventions implemented for health services continuity and interviews with key informants at various levels of the health system, including national to the community level." Several sub-part deals with document review, data collection and analysis, and ethical consideration. Obtaining information from "good informed" people are usually fruitful. But there is a lack of such information as choosing systems and intrinsic distribution among them regarding the position in the governance system.
Response: This is a valid comment from the reviewer. We have added some justification for the choice of key informants: Key informants at national level provided perspectives relating to the strategic, administrative and policy interventions to maintain service delivery. Key informants at district and health facility level provided perspectives related to the the operational adjustments to maintain service delivery (Lines 194-197).
Comment: The part called "Results" is nothing more than detailed information on undertakings implemented in the health care system to maintain its work. The authors called it "continuity ".It is possible, and let it be. But some words of what it means are necessary. There is a lack of continuity definition, as the authors in this study understand it.
Response: We agree with this observation and the earlier comments about the need to define continuity. In this paper, we aimed to describe the interventions to maintain the delivery of services. This definition has been included in the methods section of the paper (Lines 158-166).
Comment: In this part are also a few statements from the different people. It is needless. It would probably be relevant in standard journals or magazines but not scientific ones. It is not a problem whether it is a qualitative study or not, but simply, it is not the place for such stories.
Response: We thank the reviewer for pointing out this comment. The presentation of quotes requires a careful selection of statements that illustrate ideas, illuminate experience and add to the appreciation of the intended message as has been noted by Sandelowski et al.
In addition, we note that other publications in the IJERPH have used quotes to clarify claims and support research findings. In the paper by Drabwell et al, on alcohol use after bereavement, and published in the IJERPH, the authors used several quotes to illustrate the use of alcohol and drugs as a coping strategy among people who experience loss:
“For about a month I started binge drinking, as well as upping my dose of antidepressants without medical advice” (21-year-old female, 1 year since uncle/aunt died)
“In the first month following his death I was drinking excessively and smoking marijuana a lot” (22-year-old male, less than a year since death of close friend)
Another example is that in our paper, we reported that one of the interventions for maintaining service delivery was the provision of financial and non-financial incentives to health workers. We continue to provide an example that transport was a non-financial incentive for health workers to get to health facilities to continue providing services. The quote further illustrates other ways in which health workers were able to move – due to the non-financial incentive – to provide services to patients:
Provision of financial and non-financial incentives to health workers. Furthermore, the MoH adopted strategies that enabled support to health workers to continue providing health services which included provision of financial and non-financial incentives such as accommodation and transport. At the peak of movement restrictions, health workers were transported to health facilities that were understaffed to fill human resource gaps
This was noted by key informants as follows:
“…we would move the doctor to do caesarean section especially if the patient was in a facility where the theatre was operational, rather than bring them to the regional referral hospital…. so we take the doctor there and bring them back when they are done.” KII One, District Health Officer
While we agree with the reviewer about quotes being needless if carelessly presented, the quotes provided in the paper were carefully chosen to deepen the understanding of a particular intervention as a measure to maintain service delivery.
Comment: Part four is called " discussion" and contains further information about undertakings made by the authorities to maintain the system in motion. It should stress the author's understanding that this study doesn't provide information on the effectiveness of implemented changes in the health care system. Further investigations should take into consideration the problem of efficiency as well.
Response: This is a pertinent comment from the reviewer and we did acknowledge the limitation of the fact that we do not report on the effectiveness of the interventions implemented. We have improved the discussion by citing related literature on interventions implemented elsewhere, and their effectiveness (Lines 592-600).
Comment: A few comments on the last part of the paper called "Conclusions."
The authors write, "What is evident from the study is that the foundational systems build up in stable times mitigated potential and severe damage in time of crisis". Any doubts? No. But it is general knowledge. They are not coming from this very study. Also, further conclusions are coming, like Deus ex machina. The whole paper is devoted to showing different actions to have the system in motion. So, what could be the conclusions? That so and so were introduced because of the pandemic of COVID-19. Period, Therefore, the better title for this part of the paper would be "summary."
Response: We agree with the reviewer that the importance of investing in health systems is a known fact and a preamble to the main conclusions of the paper which are
- The importance of implementing innovative strategies to ensure uninterrupted access to services during emergencies
- The importance of establishing specific coordination structures for service maintenance during emergencies
- That some strategies that were implemented during the COVID-19 pandemic can improve service delivery beyond the pandemic, while others need to be assessed for clinical utility for use outside of emergency settings
Comment: Summary and proposals
- Health systems are facing the most severe health crisis in the century.
- The priority of public health authorities is to protect the health of the .people by guaranteeing the availability of affordable health care.
- In many cases, the possibility of rendering "normal" health services in a pandemic environment was challenging. Hence special attention should be taken into consideration regarding this part of health care system activity. It could be labelled as maintaining the continuity of care.
- Relevant authorities in Uganda have undertaken many actions to do this. This paper is devoted to showing these actions.
- The subject of the paper is significant and should be edited.
Response: We thank the reviewer for this summary which is exactly what we wanted to communicate in the manuscript. We believe that the comments raised, and the edits and changes made have improved the paper significantly to mirror the summary as outlined by the reviewer in this comment.
Comment: But beforehand, a broad reshaping of the paper is necessary.
f.I would like to offer the following structure of the paper:
- Introduction.A few sentences on the COVID-19 pandemic and its impact on society and health care activity, including the health service system.
- Health care system in Uganda. This part should be devoted to characterising the organisation and functioning of the country's health care system, including the epidemic's impact. Here all definition problems should be resolved.
- Essential health service delivery. There is room for a detailed description of the authority's undertakings to mitigate the pandemic and have the system in motion.
- Summary
Response: We agree with the concerns about the contents of the paper and the need to reshape it. We have included a description of the health system in the methods section (lines 145-157), a definition of maintenance of essential services (lines 158-166) as well as a description of essential health services in Uganda (lines 211-220). These will help to orient the reader. Furthermore, we have cited literature in the discussion section to be broader than a summary. We have added citations of research published elsewhere about what other countries conducted and the effectiveness of these interventions (lines 562-590). For example, as in Uganda, we have cited work that showed that community health workers – if paid and motivated were effective in the maintenance of community based maternal and child health services. In addition, we have provided citations that showed that empowering patients to engage in self-management practices such as HIV self-testing has proven to be cost-effective when implemented in a setting where the prevalence of the undiagnosed population is over 3% as is the case for Uganda. In addition, research published elsewhere showed that dispensing medicines for longer periods is effective and alleviates disruptions in access to care and has the potential to include previously excluded populations. We have also included challenges with some interventions. For example, leveraging patient networks to distribute medicines was associated with privacy and stigma concerns. Using telemedicine was also limited to the urban areas surrounding Kampala.
Comment: It is necessary to consult the native speaker to consider the wording and mode of writing to make this paper "readable "
Response: We appreciate this comment. The entire paper has been re-read and re-edited to improve the sentence construction, grammar and flow.
Reviewer 2 Report
The manuscript of "Interventions for Continuity of Essential Health Service Delivery during the COVID-19 Response in Uganda, between 3 March 2020 and April 2021" covers a very important topic in health system research, and the readers of IJERPH would benefit from this paper if the content could be revised.
In addition to the improvement of academic English writing (only the Discussion section was well written and easy to understand), I would strongly suggest that the authors apply critical thinking skills to not only present descriptively the actions undertaken at Uganda during this period of time, but also cite current references in the literature regarding what other countries or health systems did, what worked for them and what did not, as well as what worked for Uganda and what did not, and why something worked or not worked.
I also suggest that the content could be on a specific topic, only a section of the WHO framework, to cover a very wide range of topics and every part of the framework made the manuscript a just descriptive paper.
Author Response
Dear Dr. Cecilia Xing
We thank the reviewers for expeditiously reviewing our paper and giving us valuable comments to help us improve its quality. Below this message are point by point responses to all the comments that were raised by each of the reviewers. We thank you for considering our paper in your esteemed journal. I have submitted the revised version of the paper on behalf of my co-authors. All changes to the original version of the paper have been left in track-changes so that the reviewers can easily see them.
Regards,
Steven Ndugwa Kabwama MSc Public Health
Reviewer #2
Comment: The manuscript of "Interventions for Continuity of Essential Health Service Delivery during the COVID-19 Response in Uganda, between 3 March 2020 and April 2021" covers a very important topic in health system research, and the readers of IJERPH would benefit from this paper if the content could be revised.
Response: We thank the reviewer for this comment and we have revised the paper taking into consideration the comments raised by both reviewers to improve the overall quality of the paper.
Comment: In addition to the improvement of academic English writing (only the Discussion section was well written and easy to understand), I would strongly suggest that the authors apply critical thinking skills to not only present descriptively the actions undertaken at Uganda during this period of time, but also cite current references in the literature regarding what other countries or health systems did, what worked for them and what did not, as well as what worked for Uganda and what did not, and why something worked or not worked.
Response: The comment to improve the grammar and sentence construction has been raised by the two reviewers and is well noted. The suggestion to compare findings from other researchers is an insightful one. We have added citations of research published elsewhere about what other countries conducted and the effectiveness of these interventions (Lines 532-590). For example, as in Uganda, we have cited work that showed that community health workers – if paid and motivated were effective in the maintenance of community based maternal and child health services. In addition, we have provided citations that showed that empowering patients to engage in self-management practices such as HIV self-testing has proven to be cost-effective when implemented in a setting where the prevalence of the undiagnosed population is over 3% as is the case for Uganda. In addition, research published elsewhere showed that dispensing medicines for longer periods is effective and alleviates disruptions in access to care and has the potential to include previously excluded populations. We have also included challenges with some interventions. For example, leveraging patient networks to distribute medicines was associated with privacy and stigma concerns. Using telemedicine was also limited to the urban areas surrounding Kampala.
Comment: I also suggest that the content could be on a specific topic, only a section of the WHO framework, to cover a very wide range of topics and every part of the framework made the manuscript a just descriptive paper.
Response: This is an insightful suggestion by the reviewer. For example, a description of the workforce modifications to maintain services for HIV/AIDS, TB, Malaria, Diabetes, Measles etc. However, we chose to approach the research question more broadly starting from the health system as a whole and not its capacity to maintain specific health services. As such, it was important to highlight interventions across each of the health system building blocks, using specific interventions for specific desease programs as examples. For example, dispensing medicines for multiple months was a specific example of a service delivery strategy for maintaining access to ART.
Round 2
Reviewer 2 Report
I'm now happy with the revision, and the manuscript is ready to be published.